# Novel South African Rare Actinomycete *Kribbella*
*speibonae* Strain SK5: A Prolific Producer of Hydroxamate Siderophores including New Dehydroxylated Congeners

**DOI:** 10.3390/molecules25132979

**Published:** 2020-06-29

**Authors:** Kojo Sekyi Acquah, Denzil R. Beukes, Digby F. Warner, Paul R. Meyers, Suthananda N. Sunassee, Fleurdeliz Maglangit, Hai Deng, Marcel Jaspars, David W. Gammon

**Affiliations:** 1Department of Chemistry, University of Cape Town, Rondebosch 7701, South Africa; ACQKOJ001@myuct.ac.za (K.S.A.); snsunassee@gmail.com (S.N.S.); 2School of Pharmacy, University of the Western Cape, Bellville 7535, South Africa; dbeukes@uwc.ac.za; 3SAMRC/NHLS/UCT Molecular Mycobacteriology Research Unit & DST/NRF Centre of Excellence for Biomedical TB Research, Department of Pathology, Faculty of Health Sciences, University of Cape Town, Observatory 7925, South Africa; digby.warner@uct.ac.za; 4Institute of Infectious Disease and Molecular Medicine, Faculty of Health Sciences, University of Cape Town, Observatory 7925, South Africa; 5Wellcome Centre for Infectious Diseases Research in Africa, University of Cape Town, Rondebosch 7701, South Africa; 6Department of Molecular and Cell Biology, University of Cape Town, Rondebosch 7701, South Africa; paul.meyers@uct.ac.za; 7College of Science, University of the Philippines Cebu, Lahug, Cebu City 6000, Philippines; 01fm16@abdn.ac.uk; 8Marine Biodiscovery Centre, Department of Chemistry, University of Aberdeen, Old Aberdeen AB24 3UE, UK; h.deng@abdn.ac.uk (H.D.); m.jaspars@abdn.ac.uk (M.J.)

**Keywords:** *Kribbella*, speibonoxamine, siderophore, hydroxamate, molecular networking, mass spectrometry

## Abstract

In this paper, we report on the chemistry of the rare South African Actinomycete *Kribbella speibonae* strain SK5, a prolific producer of hydroxamate siderophores and their congeners. Two new analogues, dehydroxylated desferrioxamines, speibonoxamine **1** and desoxy-desferrioxamine D_1_
**2**, have been isolated, together with four known hydroxamates, desferrioxamine D_1_
**3**, desferrioxamine B **4**, desoxy-nocardamine **5** and nocardamine **6**, and a diketopiperazine (DKP) **7**. The structures of **1**–**7** were characterized by the analysis of HRESIMS and 1D and 2D NMR data, as well as by comparison with the relevant literature. Three new dehydroxy desferrioxamine derivatives **8**–**10** were tentatively identified in the molecular network of *K.*
*speibonae* strain SK5 extracts, and structures were proposed based on their MS/MS fragmentation patterns. A plausible *spb* biosynthetic pathway was proposed. To the best of our knowledge, this is the first report of the isolation of desferrioxamines from the actinobacterial genus *Kribbella*.

## 1. Introduction

Minerals are essential for the growth, development, and propagation of living organisms. Iron, a crucial element, is involved in various cellular processes including oxygen metabolism, electron transfer, DNA and RNA biosynthesis, and as a catalyst in enzymatic processes where it serves as a cofactor for many enzymes [1]. Microorganisms require iron for biofilm formation, and a lack of this mineral reduces the hydrophobicity of the microbial surface, leading to restriction in biofilm formation [2]. Although the Earth is endowed with copious amounts of iron, it is mostly in an insoluble oxidized form, Fe(OH)_3_; there are insufficient amounts of the Fe(II) form to meet cellular needs. Plants and microorganisms overcome this limitation by producing Fe(III)-chelating siderophores to scavenge and accumulate iron from soil, fresh and marine water, and sediments for absorption and subsequent reduction to the required Fe(II) form [3].

Siderophores are low molecular weight compounds (200–2000 Da) produced by bacteria, fungi, and graminaceous plants under iron limiting conditions [4]. These compounds form complexes with Fe(III) in the extracellular environment, which are then taken up into the cell via specific high-affinity uptake proteins. Iron, in its soluble Fe(II) form, is subsequently liberated from the siderophore via a redox-mediated process [3]. Siderophores have agricultural, biological control, environmental and medicinal applications. In agriculture, they increase soil fertility by making Fe(II) readily available and aid in nitrogen fixation [3]. Siderophores from nonpathogenic microorganisms compete for iron with those produced by plant and fish fungal pathogens in soil and water habitats, respectively, thereby serving as biological control agents [4]. Since siderophores chelate other metal cations (divalent, trivalent, and actinides) in soil and water, they tend to reduce the level of metal contamination in the environment [5]. The siderophore, desferrioxamine B, isolated from several *Streptomyces* species, is used to remove excess iron in patients suffering from iron overload [6]. Furthermore, siderophores linked to antibiotics show a higher antibacterial potency compared to normal antibiotics due to an enhanced uptake using the siderophore-mediated iron active transport. Examples of siderophore-enhanced antibiotics are the natural albomycin (thioribosyl pyrimidine antibiotic and tri-hydroxamate siderophore) and the synthetic, FDA-approved cefiderocol (cephalosporin antibiotic and catechol siderophore) [6]. Siderophores are also being explored in the synthesis of sideromycin (siderophore linked to antibiotics) for antibiotic drug discovery and have been explored in a “Trojan horse” approach to a novel anti-tuberculosis agent [7,8].

There are over 500 reported siderophores belonging to a number of different structural classes, about 270 of which have been structurally characterized [3,9]. Siderophores are grouped into four classes, namely hydroxamates, catecholates, carboxylates, and siderophores with mixed ligands, based on characteristic functional groups [10,11]. The hydroxamates are typically made of *N*-hydroxy-*N*-succinyl-cadaverine units and normally use three pairs of N-OH and C=O to coordinate with iron in an octahedral geometry [12]. Desferrioxamines and ferrioxamines are good examples, with over 20 known analogues identified to date, which have been mainly characterized by nuclear magnetic resonance (NMR) and extensive mass spectrometric (MS) methods [13].

Species of the genus *Kribbella* are classified as rare actinomycetes since they are less frequently isolated than species of other actinomycete genera, especially *Streptomyces*, although they may not be rare in the environment [14]. *Kribbella* strains are rich sources of novel secondary metabolites; for example, the antifungal alkyl glyceryl ethers, kribelloside A–D, which are produced by *Kribbella* strain MI481-42F6, which was isolated from a soil sample collected in Japan [15]. Among the *Kribbella* species isolated, 31 have currently been fully characterized [16]. However, very few *Kribbella* strains have been investigated for their metabolite profiles and/or isolation of secondary metabolites. 

Our research into South African bacterial strains for natural product molecules with antimycobacterial properties has led to the isolation of the rare actinomycete *Kribbella speibonae* strain SK5, which exhibited an antimycobacterial activity against *Mycobacterium aurum* strain A+ [17]. Herein, we report the secondary metabolites isolated from *K. speibonae* strain SK5, including two new desferrioxamines, speibonoxamine **1** and desoxy-desferrioxamine D_1_
**2**, four known hydroxamates **2**–**6** and a diketopiperazine (DKP) **7**. Although siderophores are ubiquitous among metabolites produced by actinomycete strains [3], especially under iron-deficient conditions, this is the first report of the isolation of siderophores from an actinobacterium of the genus *Kribbella*.

## 2. Results and Discussion

The *K. speibonae* strain SK5 was isolated from a topsoil sample collected from Stellenbosch in the Western Cape Province of South Africa [17]. The strain was grown in an International *Streptomyces* Project medium 2 (ISP2) broth and an Amberlite XAD 16N resin was added after 14 days of incubation at 30 °C with constant shaking. Organic solvents, methanol (MeOH), ethyl acetate (EtOAc), and dichloromethane (CH_2_Cl_2_), were used sequentially to extract the organics from the combined resin and culture broth. Then, the MeOH, EtOAc, and CH_2_Cl_2_ extracts were subjected to a separate high-pressure liquid chromatography-diode array detection, high-resolution electrospray mass spectrometry (HPLC-DAD/HRESIMS) analyses for chemical profiling (Appendix A). Further MS/MS and Global Natural Product Social (GNPS) molecular network analyses of the extracts revealed the presence of several siderophores and DKPs, some of which have not been reported previously (Appendix A). The MeOH, EtOAc, and CH_2_Cl_2_ extracts were combined and fractionated using a series of purification steps, including a modified Kupchan method [18], solid phase extraction (SPE), and HPLC to yield two new siderophores, speibonoxamine **1** and desoxy-desferrioxamine D_1_
**2**, and four known hydroxamates, **3**–**6** and a DKP **7** (Figure 1). 

### 2.1. Structure Elucidation

The structures of the known compounds, desferrioxamine D_1_
**3**, desferrioxamine B **4**, desoxynocardamine **5**, and nocardamine **6** were elucidated by comparison of the HRESIMS and NMR spectra with those reported in the literature (Appendix A) [11,19,20,21,22]. Compounds **3**–**6** belong to the hydroxamate class of siderophores [3]. The HRESIMS and NMR spectra (Appendix A) of compound **7** matched the reported diketopiperazine (DKP), hexahydro-3-((4-hydroxyphenyl)methyl)-pyrrolo[1,2-a]pyrazine-1,4-dione [20]. 

The molecular formula of compound **1,** isolated as a colourless amorphous solid, was deduced as C_27_H_50_N_6_O_6_ by HRESIMS (observed [M + H]^+^ = 555.3857; calculated [M + H]^+^ = 555.3870; ∆ = 1.0 ppm), indicating six degrees of unsaturation (Appendix A). 

The ^1^H-NMR of **1** in DMSO-*d_6_* showed only six signals (Appendix A), including four methylenes (δ_H_ 3.00, 2.27, 1.36, 1.22), one methyl (δ_H_ 1.77), and one NH (δ_H_ 7.77) with integrals of 12, 8, 12, 6, 6, and 6, respectively. The number of carbon atoms observed in the HRESIMS was five times higher than the number of signals observed in the ^13^C-NMR spectrum (δ_C_ 171.7, 169.4, 38.9, 31.4, 29.3, 24.3). These results suggested that compound **1** had a symmetrical structure and/or repeating motifs. Analysis of the ^1^H-^1^H COSY spectrum together with integrals of the signals in the ^1^H-NMR spectrum revealed two main spin systems, one of which consisted of three repeating motifs H-3 through H-7, H3′ through H-7′, and H-3” through H-7”, and the other comprised two repeating motifs H-9 through H-10 and H-9′ through H-10′. Careful examination of the 1D NMR data and 2D NMR data suggested that the symmetrical structure of **1** consists of two succinyls, two acyl cadaverine (AC) units, and one cadaverine moiety, characteristic of a hydroxamate siderophore. The heteronuclear multiple bond correlations (HMBC) from H-7 (δ_H_ 3.00) to C-8 (δ_C_ 171.7) and H-3” (δ_H_ 3.00) to C-11′ (δ_C_ 171.7) established the connectivity of the AC subunits to the succinyl groups. Furthermore, the cross peaks from H-3′ (δ_H_ 3.00) to C-11 (δ_C_ 171.7) and H-7′ (δ_H_ 3.00) to C-8′ (δ_C_ 171.7) established the attachment of the cadaverine moiety to the rest of the molecule. 

The final structural analysis of **1** was confirmed by comparison with the spectroscopic data reported for desferrioxamine D_1_ (**3**) in the literature [21], which differs from **1** in the presence of *N*-hydroxy groups in the structure. Therefore, compound **1** represents a new non-hydroxylated desferrioxamine, which was named speibonoxamine after the producing organism, *Kribbella speibonae* strain SK5. This is the first report of a non-hydroxylated desferrioxamine. Compound **1** may not be efficient in sequestering iron from the environment because of the lack of the hydroxamate moiety. 

The molecular formula of compound **2**, also isolated as a colourless amorphous solid, was established as C_27_H_50_N_6_O_8_ by HRESIMS (observed [M + H]^+^ = 587.3754; calculated [M + H]^+^ = 587.3768; ∆ = −2.1 ppm), indicating six degrees of unsaturation (Appendix A). 

The mass of compound **2** was 32 mass units greater than **1** indicative of the presence of two additional oxygen atoms in the structure. Analysis of the ^1^H-NMR data of **2** (Appendix A) revealed chemical shifts similar to those of **1**, except for signals at δ_H_ 1.50 (H-6 and H-6′) and 3.45 (H-7 and H-7′) as the most distinguishable change. The observed proton (δ_H_ 3.45) and carbon (δ_C_ 47.6) chemical shifts in **2** were more downfield than in **1** (δ_H_ 3.00, δ_C_ 38.9), indicating that the methylenes C-7 and C-7′ were deshielded by the adjacent *N*-hydroxy groups in **2**. Likewise, the methylene signals at (δ_H_ 1.50, δ_C_ 26.5) were downfield in **2** compared to that in **1** (δ_H_ 1.36, δ_C_ 29.3) because of the attachment of C-6 and C-6′ to the deshielded C-7 and C-7′ in **2**, respectively. Compound **2** was linked to desoxynocardamine (**5**) and desferrioxamine D_1_ (**3**) in the GNPS molecular network with a mass difference of 2 and 16 Da, respectively (Appendix A) indicating that **2** is an acyclic analogue of **5** or dehydroxylated analogue of **3**. The structure of **2** was determined by analyses of the MS/MS fragmentation data together with a comprehensive interpretation of the 1D and 2D NMR data of **2** and comparison with literature data. Compound **2** is a new dehydroxy analogue of compound **3** and hence has been assigned the name desoxy-desferrioxamine D_1_ [21]. 

### 2.2. Molecular Network Analysis

Dereplication of the metabolites produced by the *K. speibonae* strain SK5 was pivotal in the detection and isolation of the new compounds. Spectrometric data of the metabolites in the crude extract were searched against the natural product database, AntiBase (2017), with subsequent analysis of the data on the GNPS molecular networking platform [23], which grouped the compounds into clusters based on the similarity in MS/MS fragmentation patterns. Several known siderophores and DKPs were identified in the molecular network, including desferrioxamine H **11**, ferrioxamine B **12**, ferrioxamine E **13**, ferrioxamine D_1_
**14**, arthrobactin **15** and the DKP, hexahydro-3-(phenylmethyl)-pyrrolo[1,2-a]pyrazine-1,4-dione **16** (Appendix A) [13,19]. Furthermore, three new dehydroxylated siderophores **8**–**10** were putatively identified in the molecular network, in addition to the new compounds, speibonoxamine **1** and desoxy-desferrioxamine D_1_
**2**, isolated and reported in this study.

Compound **8** was linked to compound **2** when default parameters were used to generate the GNPS molecular network (Appendix A). Compound **8** (*m*/*z* 571.3807 [M + H]^+^, C_27_H_50_N_6_O_7_) was identified as the dehydroxy analogue of **2** (C_27_H_50_N_6_O_8_) and the hydroxyl analogue of **1** (C_27_H_50_N_6_O_6_), as they have a mass difference of 16 Da and the same double bond equivalent (DBE) of six (Figure 2 and Appendix A). Inspection of the fragment ions of compounds **1**, **2**, and **8** was very useful in assigning the position of the hydroxyl group, especially fragment ion 411.2596 (C_20_H_35_O_5_N_4_), which was present in **1** and **8**, but absent in **2** (Appendix A). Although the new compound **8** could not be isolated due to a paucity of material, its structure was proposed based on the MS/MS fragmentation data analysis and was assigned the name *di*desoxy-desferrioxamine D_1_ (Figure 2). 

Compound **9** (*m*/*z* 545.3654 [M + H]^+^, C_25_H_48_N_6_O_7_) was linked to desferrioxamine B **4** in the GNPS molecular network and showed a mass difference of 16 Da (Figure 2 and Appendix A). Compound **10** (*m*/*z* 529.3702 [M + H]^+^, C_25_H_48_N_6_O_6_) was linked to **9** and had a mass difference of 16 Da. The MS/MS fragmentation patterns of **4**, **9**, and **10** were used to predict the structures of **9** and **10** (Appendix A), and they were named as desoxy-desferrioxamine B and *di*desoxy-desferrioxamine B, respectively.

The results suggest that the *K. speibonae* strain SK5 is a prolific producer not only of hydroxamate siderophores, but also dehydroxylated and non-hydroxylated desferrioxamines. Although the presence of siderophores has been previously identified in the genomes of *Kribbella* species [17,24], to our knowledge this is the first report of the isolation of hydroxamates from this genus. 

### 2.3. Proposed Biosynthetic Pathway

Biosynthesis of the desferrioxamine class of hydroxamate siderophores has been elucidated in *Streptomyces* and is mediated by nonribosomal peptide synthetase-independent siderophore (NIS) synthetases [9,10,25,26]. Every NIS biosynthetic pathway identified to date contains at least one synthetase with a high sequence similarity to IucA/IucC and such synthetases have thus become the characteristic feature of these pathways [10]. In silico analysis of the annotated genome of the *K. speibonae* strain SK5 (GenBank accession number: SJJY00000000) identified the biosynthetic gene cluster (BGC) encoding the IucA/IucC synthetase that is likely responsible for producing **1**–**6** (Figure 3A, Table 1). 

The desferrioxamine (des) BGC in *Streptomyces coelicolor* strain M145 was proposed to comprise four genes, desABCD [25]. The spbA and spbB genes in the *K. speibonae* strain SK5 encode enzymes having a high sequence similarity to pyridoxal 5′-phosphate (PLP)-dependent decarboxylase and lysine-6-monooxygenase in Streptomyces strains, respectively [27]. These two enzymes are proposed to catalyze the decarboxylation of lysine **17** to form cadaverine **18** followed by hydroxylation to generate *N*-hydroxy-cadaverine **19** (Figure 3B). The acyltransferase, SpbC, is likely to catalyze the *N*-acylation of **19** [9,25], which then undergoes ATP-dependent oligomerization and macrocylization catalyzed by SpbD and SpbE to produce the *N*-hydroxy-containing hydroxamate desferrioxamines **2**–**6** [28]. 

The isolation of speibonoxamine **1** from *K. speibonae* strain SK5 suggests that the biosynthetic enzyme, *N*-acyl-CoA transferase SpbC, may display substrate promiscuity, binding not only to *N*-hydroxy-cadaverine **19** but also cadaverine **18**, the key difference between the biosynthesis of desferrioxamines and speibonoxamine **1**. It is proposed that the common decarboxylation intermediate, cadaverine **18,** undergoes spontaneous SpbC-catalyzed acylation, followed by SpbD- and SpbE-catalyzed oligomerization of two molecules of succinate, two molecules of *N*-acetyl-cadaverine, and one molecule of cadaverine to generate **1**. 

## 3. Materials and Methods

### 3.1. General Experimental Procedures

NMR spectra were obtained on a BRUKER Ascend 600 (Bruker, Billerica, MA, USA) Prodigy cryoprobe at 600 and 150 MHz for ^1^H and ^13^C nuclei, respectively. DMSO-*d*_6_ (δ_H_ 2.50, δ_C_ 39.7), CD_3_OD (δ_H_ 3.30, δ_C_ 49.0), and CDCl_3_ (δ_H_ 7.25, δ_C_ 77.00) were used for preparing samples for NMR experiments. High resolution mass spectrometric data were obtained using a Thermo Instruments MS system (LTQ XL/LTQ Orbitrap Discovery, Thermo Scientific, Bremen, Germany) coupled to a Thermo Instruments HPLC system (Accela PDA detector, Accela PDA autosampler, and Accela pump). The MS was run in a positive high-resolution mode (60,000) and MS/MS at a resolution of 7500 and a low-resolution negative mode. The following conditions were used: Capillary voltage 45 V, capillary temperature 260 °C, auxiliary gas flow rate 10–20 arbitrary units, sheath gas flow rate 40–50 arbitrary units, spray voltage 4.5 kV, mass range 100–2000 amu (maximum resolution 30,000). HPLC separations were carried out using a reverse phase C18 ACE 10 µM 10 × 250 mm column connected to an Agilent Technologies 1200 series HPLC system equipped with an Agilent Technologies 1200 series quad pump and Agilent Technologies 1200 series DAD. The amberlite XAD 16N resin was obtained from Sigma-Aldrich, Johannesburg, South Africa. All solvents used throughout were of a HPLC-grade and purchased from both Merck and Sigma-Aldrich (Johannesburg, South Africa).

### 3.2. Isolation and Characterization of the Strain

The *K. speibonae* strain SK5 was isolated from a topsoil sample collected from the town of Stellenbosch in the Western Cape Province of South Africa using a newly developed *Kribbella*-selective medium [17].

### 3.3. Fermentation

A liquid stock culture of the *K. speibonae* strain SK5 was inoculated into a 15 mL International *Streptomyces* Project medium 2 (ISP2) broth (yeast extract 4 g, malt extract 10 g, glucose 4 g, in 1 L H_2_O, pH 7.3 [29]) in a 250 mL Erlenmeyer flask and incubated for five days at 30 °C with shaking. Then, the entire culture was inoculated into a 50 mL ISP2 broth in a 500 mL Erlenmeyer flask and incubated at 30 °C with shaking for four days. This culture was subsequently split into three parts, which were used as the inocula for 3 × 100 mL ISP2 broths in 1000 mL Erlenmeyer flasks and incubated at 30 °C with shaking for four days. Then, each 100 mL culture was inoculated into a 1000 mL ISP2 broth in a 5000 mL Erlenmeyer flask and incubated at 30 °C with shaking. After 14 days of incubation, the Amberlite XAD 16N resin (50 g/L) was added under sterile conditions to each flask and further incubated for 6 h at 30 °C with shaking. Subsequently, the cultures were harvested and filtered under pressure using a piece of glass wool placed in a Buchner funnel. The filtrate was partitioned in a separating funnel with an equal volume of ethyl acetate. Then, the cell mass mixed with the Amberlite XAD 16N resin (containing the adsorbed organics) was extracted sequentially with MeOH (3×), then EtOAc (3×), and finally, CH_2_Cl_2_ (3×). All the organic extracts were concentrated under a reduced pressure to give 3.43 g of the MeOH extract, 0.51 g of the EtOAc extract, and 0.19 g of the CH_2_Cl_2_ extract. The extracts were subjected to HPLC-DAD/HRESIMS analyses. 

### 3.4. HPLC-DAD/HRESIMS Analyses

Chemical profiling of the MeOH, EtOAc, and CH_2_Cl_2_ extracts was performed by HPLC-DAD/HRESIMS analyses. Each extract (10 µL, 0.1 mg/mL in MeOH) was injected and chromatographically separated on a C18 reverse-phase HPLC column (ACE 10 µM 10 × 250 mm) using a linear gradient from 95% solvent A (0.1% formic acid in water) to 100% solvent B (0.1% formic acid in acetonitrile) for 25 min, followed by 100% B for 5 min at a flow rate of 1.5 mL/min. The DAD of the HPLC was scanned from 200–400 nm. The MS was run in a positive high-resolution mode (60,000) and MS/MS at a resolution of 7500 and a low-resolution negative mode. The Xcalibur software was used to process the raw data. The exact mass and molecular formula of each peak was entered as a single query in the commercially available AntiBase (2017) Natural Compound Identifier (https://www.wiley.com/en-us/AntiBase%3A+The+Natural+Compound+Identifier-p-9783527343591) to ascertain whether the data matched any compound in the database. The HPLC-DAD/HRESIMS profiles of the three extracts showed similar chemical profiles, hence they were combined to obtain 4.13 g. 

### 3.5. Fractionation, Isolation, and Purification of Compounds

The combined crude extract (4.13 g) was suspended in 50 mL of distilled water and extracted with equal volumes of CH_2_Cl_2_ (three times). Then, the water layer was partitioned with the same volume of n-butanol (three times). The n-butanol layer (332.2 mg) was concentrated under a reduced pressure and fractionated on a C18 solid phase extraction (SPE) column using a stepwise elution of solvent mixtures of decreasing polarity (8 column volume/solvent mixture): 100% water (SPE1), 12.5% MeOH (SPE2), 25% MeOH (SPE3), 50% MeOH (SPE4), 100% MeOH (SPE5), and 100% MeOH containing 0.05% trifluoroacetic acid (SPE6). All fractions were subjected to the HPLC/HRESIMS analysis. 

Fractions SPE2-4 were further purified by the reverse-phase HPLC analysis (C18, linear gradient 100% H_2_O to 100% MeOH in 45 min, flow rate 1.5 mL/min). SPE2 yielded compounds **6** (19.3 mg) and **7** (0.8 mg), SPE3 yielded compounds **1** (1.2 mg), **2** (0.6 mg), **3** (1.2 mg), and **5** (0.8 mg), and SPE4 yielded **4** (0.5 mg). 

Speibonoxamine **1**: Colorless amorphous solid; for ^1^H, ^13^C NMR data, see Table 2; HRESIMS (positive mode) *m*/*z* 555.3857 [M + H]^+^ and 577.3672 [M + Na]^+^ Δ 1.004; calcd. for C_27_H_50_N_6_O_6_. 

Desoxy-desferrioxamine D_1_
**2**: Colorless amorphous solid; ^1^H, ^13^C NMR data, see Table 2; HRESIMS (positive mode) *m*/*z* = 587.3754 [M + H]^+^ and 609.3573 [M + Na]^+^ Δ −2.092 ppm; calcd. for C_27_H_50_N_6_O_8_.

Desferrioxamine D_1_
**3**: Colorless amorphous solid; ^1^H NMR data, see Appendix A; HRESIMS (positive mode) *m*/*z* = 603.3708 [M + H]^+^ and 625.3525 [M + Na]^+^ Δ −0.685 ppm; calcd. for C_27_H_50_N_6_O_9_.

Desferrioxamine B **4**: Colorless amorphous solid; ^1^H NMR data, see Appendix A; HRESIMS (positive mode) *m*/*z* = 561.3602 [M + H]^+^ and 583.3415 [M + Na]^+^ Δ −1.049 ppm; calcd. for C_25_H_48_N_6_O_8_.

Desoxynocardamine **5**: Colorless amorphous solid; ^1^H NMR data, see Appendix A; HRESIMS (positive mode) *m*/*z* = 585.3602 [M + H]^+^ and 607.3418 [M + Na]^+^ Δ −0.391 ppm; calcd. for C_27_H_48_N_6_O_8_.

Nocardamine **6**: Colorless amorphous solid; ^1^H, ^13^C NMR data, see Appendix A; HRESIMS (positive mode) *m*/*z* = 601.3552 [M + H]^+^ and 623.3359 [M + Na]^+^ Δ −0.161 ppm; calcd. for C_27_H_48_N_6_O_9_.

Hexahydro-3-((4-hydroxyphenyl)methyl)-pyrrolo[1,2-a]pyrazine-1,4-dione **7**: Colorless amorphous solid; ^1^H NMR data, see Appendix A; HRESIMS (positive mode) *m*/*z* = 261.1237 [M + H]^+^ and 283.1655 [M + Na]^+^ Δ 1.459 ppm; calcd. for C_14_H_16_N_2_O_3_.

### 3.6. Molecular Networking

Raw data obtained from the LC-MS/MS system were converted to a mzXML format using the ProteoWizard tool MSconvert (version 3.0.10051, Vanderbilt University, Nashville, TN, USA) [30]. All mzXML data were uploaded to the Global Natural Products Social (GNPS) molecular networking (MN) webserver3 (http://gnps.ucsd.edu) and analyzed using the MN workflow [23]. The data were filtered by removing all MS/MS fragment ions within +/− 17 Da of the precursor *m*/*z*. MS/MS spectra were window filtered by choosing only the top six fragment ions in the +/− 50 Da window throughout the spectrum. The precursor ion mass tolerance was set to 0.02 Da and a MS/MS fragment ion tolerance of 0.02 Da. Then, a network was created where edges were filtered to have a cosine score above 0.7 and more than three matched peaks. Further, edges between two nodes were kept in the network if and only if each of the nodes appeared in each other’s respective top 10 most similar nodes. Finally, the maximum size of a molecular family was set to 100, and the lowest scoring edges were removed from molecular families until the molecular family size was below this threshold. The spectra in the network were then searched against GNPS’ spectral libraries. The library spectra were filtered in the same manner as the input data. All matches kept between the network spectra and library spectra were required to have a score above 0.7 and at least three matched peaks. The output of the molecular network was visualized using the Cytoscape version 3.7.2 (https://cytoscape.org/) [31] and displayed using the settings “preferred layout” with “directed” style. The nodes (compounds) originating from the culture medium and solvent control (MeOH) were excluded from the original network to enable visualization of only the *K. speibonae* strain SK5 metabolites derived from the MeOH, EtOAC, and CH_2_Cl_2_ extracts of the cultures.

## 4. Conclusions 

The *K. speibonae* strain SK5 is a prolific producer of hydroxamate (desferrioxamine) siderophores and their dehydroxy analogues. Two new desferrioxamines, speibonoxamine and desoxy-desferrioxamine D_1_, four known hydroxamates, and a DKP were isolated from *K. speibonae* strain SK5. Furthermore, several new and known siderophores were identified in the *K. speibonae* strain SK5 extracts, and their plausible structures were determined by the MS/MS fragmentation analysis. This is the first report of siderophores isolated from the genus *Kribbella*. The proposed speibonoxamine pathway (Figure 3B) suggests a biosynthetic machinery distinct from that reported for desferrioxamine biosynthesis.

## Figures and Tables

**Figure 1 molecules-25-02979-f001:**
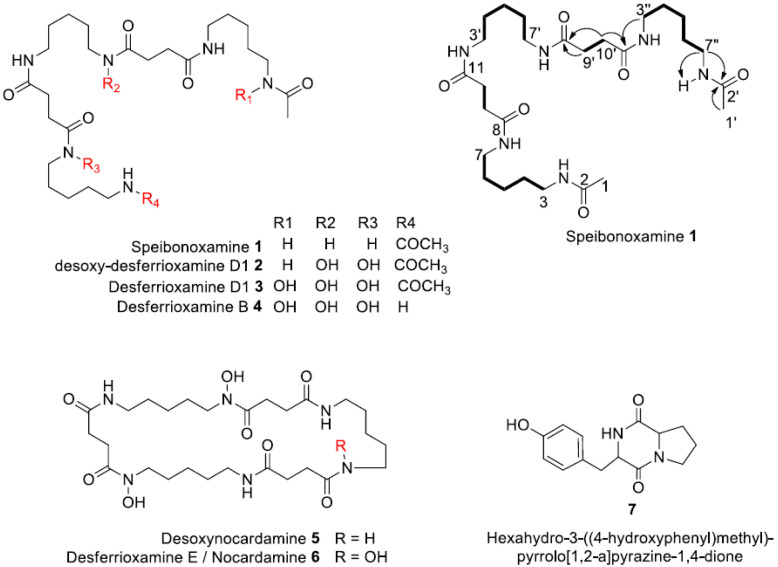
Structures of isolated metabolites **1**–**7** from the *Kribbella speibonae* strain SK5, including depiction of Speibonoxamine **1** with COSY (**―**) and key heteronuclear multiple bond correlations (HMBC) (→) correlations.

**Figure 2 molecules-25-02979-f002:**
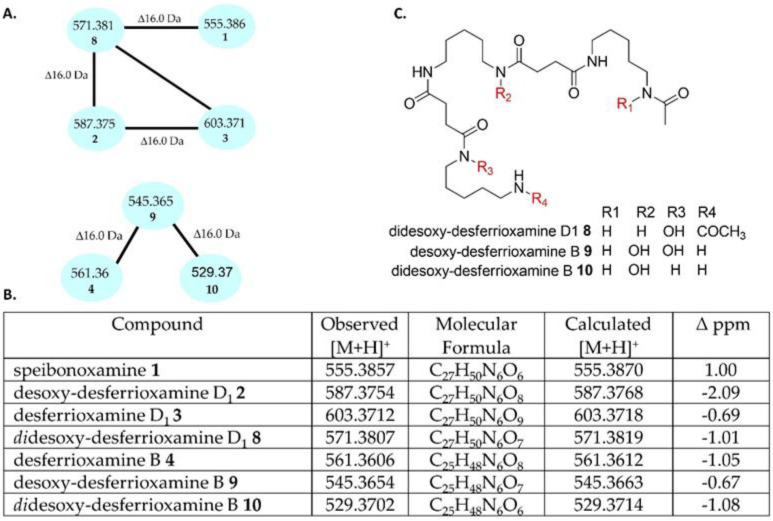
(**A**) Specific subnetwork analysis of the *K. speibonae* strain SK5; (**B**) putative dehydroxylated desferrioxamine analogues D_1_
**8** and B **9**–**10**; (**C**) structures of plausible desferrioxamines **8**–**10** determined by the MS/MS fragmentation pattern.

**Figure 3 molecules-25-02979-f003:**
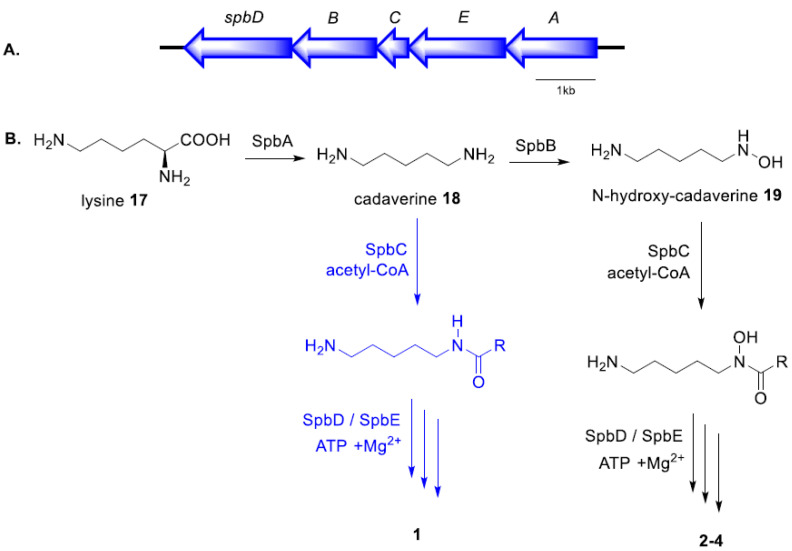
(**A**) Speibonoxamine (*spb*) biosynthetic gene cluster in the *K. speibonae* strain SK5; (**B**) proposed biosynthetic pathway of **1**–**4.**

**Table 1 molecules-25-02979-t001:** Deduced functions of open reading frames (ORFs) in *spb* biosynthetic gene cluster (BGC).

Protein	Annotated Function	*Streptomyces* Homologue% Identity/% Similarity	Amino Acid Length
SpbD	IucA/IucC synthetase	74%/83%	565
SpbB	Lysine-6-monooxygenase	80%/88%	418
SpbC	Acyltransferase	67%/76%	156
SpbE	Siderophore synthetase	78%/84%	546
SpbA	PLP-dependent decarboxylase	86%/93%	495

**Table 2 molecules-25-02979-t002:** ^1^H and ^13^C-NMR data of speibonoxamine **1** and desoxy-desferrioxamine D_1_
**2** in DMSO-*d_6._*

	Speibonoxamine 1		Desoxy-Desferrioxamine D_1_ 2	
**Position**	**^13^C**	**^1^H, Mult. (*J*, Hz)**	**^13^C**	**^1^H, Mult. (*J*, Hz)**
1, 1′	23.1, CH_3_	1.77, s	23.1, CH_3_	1.78, s
2, 2′	169.4, C	-	169.4, C	-
3, 3′, 3”	38.9, CH_2_	3.00, t (6.15)	38.9, CH_2_	3.00, m
4, 4′, 4”	29.3, CH_2_	1.36, m	29.3, CH_2_	1.38, dd (7.14, 14.24)
5, 5′, 5”	24.3, CH_2_	1.22, m	24.0, CH_2_	1.23, m
6, 6′	29.3, CH_2_	1.36, m	26.5, CH_2_	1.50, m
6”	29.3, CH_2_	1.36, m	29.3, CH_2_	1.38, dd (7.14, 14.24)
7, 7′	38.9, CH_2_	3.00, t (6.15)	47.6, CH_2_	3.45, t (6.96, 6.96)
7”	38.9, CH_2_	3.00, t (6.15)	38.9, CH_2_	3.00, m
8	171.7, C	-	172.4, C	-
8′	171.7, C	-	171.7, C	-
9	31.4, CH_2_	2.27, s	28.1, CH_2_	2.58, m
9′	31.4, CH_2_	2.27, s	29.8, CH_2_	2.40, t (6.89, 6.89)
10	31.4, CH_2_	2.27, s	31.4, CH_2_	2.27, m
10′	31.4, CH_2_	2.27, s	30.4, CH_2_	2.28, m
11	171.7, C	-	174.4, C	-
11′	171.7, C	-	171.2, C	-
NH		7.77		7.77

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
