# Peer review of "Novel South African Rare Actinomycete Kribbella speibonae Strain SK5: A Prolific Producer of Hydroxamate Siderophores Including New Dehydroxylated Congeners"

_molecules, 2020, doi:10.3390/molecules25132979_

Round 1

Reviewer 1 Report

Overall the paper is well written, the research design is appropriate and the results are clearly presented.

I have some concerns with the significance of the content. One of the new compounds identified, Speibonoxamine, lacks all the N-hydroxyl groups found in desferrioaxamines. It is pretty safe to assume that this compound lacks any siderophore function (as suggested by the authors in lines 161-162). Have the authors look at the levels of production of 1 versus 2-4? Could it just be a very minor by-product of a not very efficient biosythesis (as shown in Figure 3) without any biological role/significance?

Analogously, compounds 8-10 seem to be the result of enzyme relaxed substrate specificity and relative enzyme kinetics between SpbB, SpbC and SpbD/E. The fact that compound 8 could not be isolated due to paucity of material suggest that these compounds might be minor by-products with no biological role. Perhaps the authors could emphasize the significance of these results in case this reviewer is missing somethig?

Minor points:

Line 41: "Iron, a crucial mineral". Strictly speaking iron is an element. Either change mineral into element, or iron into any of the minerals/forms in which iron is found in earth.

Line 67: "Cefiderocol is currently in pahe III clinical trials". Please doble check this. It seems to have been approved for medical use in USA in 2019 and in the EU in 2020.

Reviewer 2 Report

The paper entitled “Novel South African rare actinomycete Kribbella speibonae strain SK5: A prolific producer of hydroxamate siderophores including new dehydroxylated congeners”, authors Kojo Sekyi Acquah, Denzil R. Beukes, Digby F. Warner, Paul R. Meyers, Suthananda N. Sunassee, Fleurdeliz Maglangit, Hai Deng, Marcel Jaspars, and David W. Gammon, is devoted to the search of novel compounds with siderophore activities. Siderophores are perspective but not well-studied biologically active compounds. Rare actinobacteria are promising candidates as producers of siderophores. Authors have isolated a new actinobacterial strain belonging to the genus Kribella. This strain is able to produce 16 different hydroxamate siderophores and their congeners, among those 5 new compounds are detected. This is the first report about synthesis of siderophores by Kribella. The paper is well-organized and clear. The background is comprehensive. A wide range of analytical methods was used in the study, such as High-Pressure Liquid Chromatography-Diode Array Detection (HPLC-DAD), High-resolution Electrospray Mass Spectrometry (HRESIMS), NMR, LC-MS/MS, and Global Natural Product Social (GNPS) molecular network analysis. Bioinformatical analysis provided with suggestions about putative ways of siderophore synthesis in rare actinobacteria. Especially I would like to make a stress that authors gave full supplementary information including the spectra and molecular networks obtained at all steps of investigation.

Specific comments

Could authors estimate and compare iron-chelating activities of hydroxamate siderophores and their congeners detected in this study?

  1. 3, lines 95-96 – K. speibonae and Streptomyces should be italicized as K. speibonae and Streptomyces.
  2. P. 3, line 108 – the subtitle “1. Structure elucidation” should be transferred to the new line below.
  3. 8, 4.2 Isolation and characterization of strain – Some information about methods and criteria used to identify this strain as Kribella speibonae is recommended to be added, as well as short information about phenotypic (biological, physiological) features of the strain.
  4. 8, lines 295 and 395 – Kribella speibonae should be reduced as K. speibonae since it is not the first appearance of organism with this Latin name in the text.

Summary. The paper is recommended for publication in the journal “Molecules” after minor revisions.

Round 2

Reviewer 1 Report

Minor points have been corrected.

The authors base the significance of the manuscript on the description of known and new siderophores in a new bacterial species. This reviewer still has doubts on the  biological importance of some of these new compounds.

Nonetheles, I support publication on its current form.